Fluorofenidone alleviates liver fibrosis by inhibiting hepatic stellate cell autophagy via the TGF-β1/Smad pathway: implications for liver cancer

Peng Xiongqun 1
Yang Huixiang 2
Tao Lijian 3
Xiao Jingni 4
Zeng Ya 1
Shen Yueming 1
Yu Xueke 1
Zhu Fei 5
Qin Jiao Qinj8429@163.com 4
1 Department of Gastroenterology, The Affiliated Changsha Central Hospital, Hengyang Medical School, University of South China , Changsha , China
2 Department of Gastroenterology, Xiangya Hospital, Central South University , Changsha , China
3 Department of Nephropathy, Xiangya Hospital, Central South University , Changsha , China
4 Department of Nephrology, The Affiliated Changsha Central Hospital, Hengyang Medical School, University of South China , Changsha , China
5 Department of General Surgery, The Affiliated Changsha Central Hospital, Hengyang Medical School, University of South China , Changsha , China
Zhan Cheng
Electronic publication date: 2023 Sep 28
Publication date: 2023
Volume: 11
Electronic Location ID: e16060
Received 2023 Jul 6; Accepted 2023 Aug 17
Copyright: ©2023 Peng et al.
Copyright year: 2023
Copyright holder: Peng et al.
License: This is an open access article distributed under the terms of the Creative Commons Attribution License, which permits unrestricted use, distribution, reproduction and adaptation in any medium and for any purpose provided that it is properly attributed. For attribution, the original author(s), title, publication source (PeerJ) and either DOI or URL of the article must be cited.
License URL: https://creativecommons.org/licenses/by/4.0/

Keywords: AKF-PD, Liver fibrosis, Autophagy, Hepatic stellate cell, TGF-β1/Smad pathway

Funding: The Changsha Municipal Natural Science Foundation kq2014021 The Scientific Research Project of Hunan Provincial Health Commission 202303037199 202203052625 The Hunan Provincial Natural Science Foundation - Youth Foundation 2020JJ5611 The Changsha Central Hospital Subject YNKY 202139 YNKY 202101 This study was funded by the Changsha Municipal Natural Science Foundation (Grant No. kq2014021), the Scientific Research Project of Hunan Provincial Health Commission (Grant No. 202303037199 and 202203052625, the Hunan Provincial Natural Science Foundation—Youth Foundation (Grant No. 2020JJ5611), and the Changsha Central Hospital Subject (Grant Nos. YNKY 202139 and YNKY 202101). The funders had no role in study design, data collection and analysis, decision to publish, or preparation of the manuscript.

==============================
Objectives

Liver fibrosis is a key stage in the progression of various chronic liver diseases to cirrhosis and liver cancer, but at present, there is no effective treatment. This study investigated the therapeutic effect of the new antifibrotic drug fluorofenidone (AKF-PD) on liver fibrosis and its related mechanism, providing implications for liver cancer.

Materials and Methods

The effects of AKF-PD on hepatic stellate cell (HSC) autophagy and extracellular matrix (ECM) expression were assessed in a carbon tetrachloride (CCl4)-induced rat liver fibrosis model. In vitro, HSC-T6 cells were transfected with Smad2 and Smad3 overexpression plasmids and treated with AKF-PD. The viability and number of autophagosomes in HSC-T6 cells were examined. The protein expression levels of Beclin-1, LC3 and P62 were examined by Western blotting. The Cancer Genome Atlas (TCGA) database was used for comprehensively analyzing the prognostic values of SMAD2 and SMAD3 in liver cancer. The correlation between SMAD2, SMAD3, and autophagy-related scores in liver cancer was explored. The drug prediction of autophagy-related scores in liver cancer was explored.

Results

AKF-PD attenuated liver injury and ECM deposition in the CCl4-induced liver fibrosis model. In vitro, the viability and number of autophagosomes in HSCs were reduced significantly by AKF-PD treatment. Meanwhile, the protein expression of FN, α-SMA, collagen III, Beclin-1 and LC3 was increased, and P62 was reduced by the overexpression of Smad2 and Smad3; however, AKF-PD reversed these effects. SMAD2 and SMAD3 were hazardous factors in liver cancer. SMAD2 and SMAD3 correlated with autophagy-related scores in liver cancer. Autophagy-related scores could predict drug response in liver cancer.

Conclusions

AKF-PD alleviates liver fibrosis by inhibiting HSC autophagy via the transforming growth factor (TGF)-β1/Smadpathway. Our study provided some implications about how liver fibrosis was connected with liver cancer by SMAD2/SMAD3 and autophagy.

Introduction

Liver fibrosis is characterized by excessive extracellular matrix (ECM) deposition in the liver, which is triggered by various kinds of chronic liver injury, including viral infection, alcohol, toxic substances, and metabolic diseases (Campana & Iredale, 2017; Higashi, Friedman & Hoshida, 2017). Liver fibrosis can be reversed at the early stage, but without efficient treatment, persistent liver fibrosis will develop into liver cirrhosis and even hepatocellular carcinoma (Zhou et al., 2021). Liver fibrosis is closely related to hepatocellular carcinoma (HCC), and the incidence of HCC caused by cirrhosis is as high as 90% (Shankaraiah et al., 2019). Statistics show that approximately 1 million people die from liver cirrhosis each year worldwide, and the trend has been increasing in recent years (Tsochatzis, Bosch & Burroughs, 2014). In addition, hepatocellular carcinoma is the fifth most common cancer worldwide and the third leading cause of cancer-related deaths (Attwa & El-Etreby, 2015). Therefore, liver fibrosis can serve as a potential therapeutic target for preventing or delaying the development of liver cancer. Currently, there is no efficient drug for treating human liver fibrosis.

The activation of hepatic stellate cells (HSCs) plays a crucial part in the progression of liver fibrosis (Higashi, Friedman & Hoshida, 2017). Under normal physiological conditions, the key feature of HSCs is to reserve and metabolize vitamins (Friedman, 2008), but after liver damage, HSCs are activated and transform into myofibroblast cells, whose markers include fibronectin (FN) and α- smooth muscle actin (α-SMA) (Tsuchida & Friedman, 2017). Moreover, activated HSCs can release transforming growth factor (TGF)-β1, which is deemed to be the most effective factor to promote fibrocytes (Hellerbrand et al., 1999). TGF-β1 can induce the activation of HSCs and the production of collagen through the TGF-β1/Smad pathway, thereby promoting the progression of liver fibrosis (Peng et al., 2019). TGF-β1/Smad signaling pathway is also particularly important for the occurrence and development of tumors (Nong et al., 2019). Abnormal activation of TGF-β/Smad signaling pathways promotes tumor metastasis in HCC (Gao et al., 2019; Liao et al., 2020; Wang et al., 2021).

Autophagy is a degradative process by which impaired organelles, long-lived proteins and invading microorganisms in the cytoplasm of eukaryotes are degraded to maintain cell homeostasis (Sir et al., 2010). To date, there have been many studies on the autophagy mechanism. The formation of autophagosomes is a characteristic manifestation of autophagy, and many markers related to autophagy have been found, including Beclin-1 and microtubule-associated protein light chain 3 (LC3) (Jia et al., 2019; Ruart et al., 2019). Meanwhile, several signalling pathways related to autophagy were also found, such as the AMPK-mTOR pathway (Jin et al., 2016) and TGF-β1/Smad pathway (Xia et al., 2020). Autophagy plays a regulatory effect in multiple liver diseases, including liver fibrosis and liver cancer (Wang et al., 2019). HSC autophagy promotes lipid droplet degradation and provides energy for inducing HSC activation (Hernández-Gea et al., 2012). Recently, it has been reported that some drugs play an anti-liver fibrosis role by inhibiting the autophagy of HSCs (Hao et al., 2016; Liu et al., 2019; Ma et al., 2019). In addition, previous research has shown that autophagy plays a role in various stages of tumor formation, occurrence, development, invasion, and metastasis (White, 2012). Ma et al. (2017) have found that TGF-β1 promotes human hepatic carcinoma cells invasion by upregulating autophagy. In 1999, Fufang Biejia Ruangan tablets (FFBJ) were the first traditional Chinese medicine authorized by the China Food and Drug Administration (CFDA) to treat liver fibrosis (Guo et al., 2004). FFBJ can attenuate liver fibrosis by inhibiting TGF-β1/Smad in a CCl4-induced fibrosis model (Yang, Fang & Lou, 2013). FFBJ can also treat liver fibrosis by preventing HSC proliferation and activation (Guo et al., 2004).

Fluorofenidone [1-(3-fluorophenyl)-5-methyl-2-(1H)-pyridone; AKF-PD] is a novel pyridone antifibrotic drug independently developed by Central South University. Compared with pirfenidone (PD), AKF-PD has been found to exhibit a lower toxicity, longer half-life and equivalent antifibrotic activity (Lou et al., 2012). AKF-PD possesses various beneficial pharmacological activities including anti-inflammation (Tu et al., 2021), anti-fibrosis (Peng et al., 2014; Tu et al., 2021), anti-oxidation (Jiang et al., 2019b), anti-apoptosis (Yang et al., 2019) and anti-necroptosis (Dai et al., 2020). Preceding researches have proven that AKF-PD has anti liver fibrosis effects in several animal models (Peng et al., 2014; Peng et al., 2019), Peng et al. (2014) found that AKF-PD can alleviate liver fibrosis by restraining HSC activation and proliferation, and it can regulate the TGF-β1/Smad and MAPK pathways to inhibit the activation of hepatic stellate cells (Peng et al., 2013; Peng et al., 2014; Peng et al., 2019). However, the relationship between AKF-PD and autophagy is not completely clear. The purpose of this research was to study whether AKF-PD alleviates liver fibrosis by restraining HSC autophagy via the TGF-β1/Smad pathway.

A recent emphasis has been on better understanding and potentially targeting the microenvironment in which liver tumors form. This is due to the broad landscape of genomic alterations and the limited therapeutic success of targeting tumor cells. The tight relationship between liver fibrosis and liver cancer is a distinctive characteristic. Fibrosis and cancer-associated fibroblasts (CAF) can influence HCC development by modulating the tumor microenvironment (Baglieri, Brenner & Kisseleva, 2019). TGF-β signaling involves all phases of the development of liver fibrosis and HCC (Fabregat & Caballero-Díaz, 2018). Activating TGF-β signaling in HSCs promotes the extracellular matrix formation (Seki & Schwabe, 2015) and promotes HCC growth (Mikula et al., 2006). Gao et al. (2019) have found that the levels of SMAD2, SMAD3, and SMAD4 in liver cancer are significantly elevated. Autophagy plays an important regulatory role in liver fibrosis and HCC,and autophagy-related gene signatures can be used as a biomarker to predict clinical outcomes of HCC (Shen & Lin, 2019; Mao et al., 2020).Owing to the close relationship between liver fibrosis and liver cancer. The prognostic values of SMAD2 and SMAD3 in liver cancer were explored. The correlation between SMAD2, SMAD3, and autophagy-related scores in liver cancer was explored. The drug prediction of autophagy-related scores in liver cancer was explored.

Materials and Methods

Liver fibrosis model and AKF-PD treatment

All animals used in this research were five- to six-week-old male Sprague–Dawley rats (Hunan SJA Laboratory Animal Co., Ltd. Changsha, China). The experimental protocol was approved by the Ethics Review Committee for Animal Experimentation of University of South China (Hengyang, China)(approval number: usc202104xs45). Animal studies were reported in compliance with the ARRIVE guidelines. Twenty-four rats were placed in eight cages and raised in controlled conditions (22 ± 2 °C, 45∼55% relative humidity, 12-h day and night cycle) and provided food and water freely during the experiment. The rats were separated into four groups (six rats/group): I : normal group; II : CCl4 group; III: CCl4+AKF-PD (240 mg/kg/day) group and IV : CCl4+ FFBJ (550 mg/kg/day) group. In Groups II, III and IV the rats were treated by intraperitoneal injection of CCl4 (#40006861 Sinopharm Chemical Reagent Co. Ltd., Beijing, China) (2 ml/kg body weight, 1:1 in olive oil, twice weekly) for 6 weeks. The rats in Group I received an intraperitoneal injection of the same dose of olive oil. The rats in Group III were intragastrically administered AKF-PD (lot No. 20190810 Haikou, China) (240 mg/kg/d once a day) (Peng et al., 2019) for 6 weeks. In Group IV, the rats were intragastrically administered FFBJ (#C0121041, Inner Mongolia Furui Medical Science Co., Ltd., China) (550 mg/kg/d once daily) (Yang, Fang & Lou, 2013) for 6 weeks. In Groups I and II , the rats were intragastrically administered 0.5% carboxymethyl cellulose sodium (CMCNa) once daily for 6 weeks. At the end of week 6, all rats were euthanized with an overdose anesthetizing of sodium pentobarbitone (90 mg/kg, intraperitoneal injection) (Moradian et al., 2022). Serum and liver samples were acquired. Serum was applied to examine indicators of liver function. Livers were rapidly harvested, rinsed in cold saline and weighed while wet. A portion of liver was fixed (room temperature) in 10% neutral-buffered formalin for histopathological staining and the remaining tissue was stored at −70 °C for Western blotting.

ALT/AST/TBIL/ALB assays

The serum levels of alanine aminotransferase (ALT), albumin (ALB), aspartate aminotransferase (AST) and Total Bilirubin (TBIL) were detected by ALT Kit (#C009-2-1, Jiancheng, China), ALB Kit (#A028-2-1, Jiancheng, China), AST Kit (#C010-2-1, Jiancheng, China) and TBIL (#C019-1-1, Jiancheng, China)following the manufacturer’s directions.

Histopathological staining

Hematoxylin and eosin (H&E), Sirius Red staining and Masson’s trichrome staining were used for histological analyses. Sirius Red staining and Masson’s trichrome staining were applied to show the degree of fibrosis. The blue area of Masson’s trichrome staining and the red area of Sirius Red staining in the image showed the distribution areas of collagen in the liver. Olympus microscope was used to randomly select 5 non-overlapping fields, percentage of positive area of fibrosis was calculated by Image-Pro Plus 8.0.

Cell culture and treatment

The rat HSC line (HSC-T6) (#BNCC337976) was provided by BeNa culture collection (Beijing, China). HSC-T6 cells were cultured in DMEM with 10% FBS and 1% penicillin/streptomycin and cultured to 60–70% confluence for 24 h at 37 °C. The cells were exposed to 2 mM AKF-PD (In the previous study, AKF-PD in 2 mM can exert the optimal effect)for 24 h after stimulation with recombinant human TGF-β1 (5 ng/ml, 24 h) (Peng et al., 2013; Peng et al., 2014; Peng et al., 2019).

Cell viability assay

HSC-T6 cells were grown in 96-well plates, followed by 2 mM AKF-PD and incubation for 0h, 24 h, 48 h and 72 h. Cell viability was analysed by Cell Counting Kit-8 (CCK-8; APExBIO, Houston, TX, USA).

Acridine orange fluorescent staining

HSC-6 cells were grown in 6-well plates for 48 h, treated with 2 mM AKF-PD for 24 h, and stained with 0.01% acridine orange (Solarbio, CA1142, China) for 30 min in dark room. The cells were examined with a fluorescence microscope (magnification, ×400) (Olympus ABX50, Tokyo, Japan). The area of red fluorescence was quantified using Image-Pro Plus 8.0.

Western blot analysis

Liver tissues and HSC-T6 cells were lysed with RIPA buffer (#ab156034, Abcam, Cambridge, UK) containing a phosphatase inhibitor and PMSF (#ab141032, Abcam, Cambridge, UK). Protein samples were denatured and separated by 10–15% SDS-polyacrylamide gel and transferred to polyvinylidene difluoride membranes (Millipore, Burlington, MA, USA). The membranes were blocked and incubated with primary antibodies. The primary antibodies were as follows: collagen I (1:1000; #ab270993, Abcam, UK), TGF-β1 (1:500; #sc-130348, Santa Cruz, CA, USA), collagen III (1:1000; #ab184993, Abcam, UK), Smad2 (1:500; #sc-393312, Santa Cruz, CA, USA), Smad3 (1:500; #sc-101154, Santa Cruz, CA, USA), p-Smad2 (1:1000; #ab280888, Abcam, Cambridge, UK), p-Smad3 (1:1000; #ab52903, Cambridge, Abcam, UK), LC3 (1:1000; #ab192890, Abcam, Cambridge, UK), Beclin-1 (1:1000; #ab207612, Abcam, Cambridge, UK), P62 (1:1000; #sc-28359, Santa Cruz, CA, USA), α-SMA (1:1000; #55135-1-AP Proteintech, Rosemont, IL, USA), fibronectin (FN) (1:500; #sc-8422, Santa Cruz, CA, USA), and GAPDH (1:1000; #ab8245, Abcam, Cambridge, UK). Subsequently, the membranes were incubated with HRP anti-rabbit IgG (1:2000; #ab288151, Abcam, Cambridge, UK). Finally, the bands were visualized using the ECL- chemiluminescent kit (Proteintech, Rosemont, IL, USA) and quantified using ImageJ software.

Transmission electron microscopy (TEM)

HSC-T6 cells were maintained in 2.5% glutaraldehyde for 12 h and fixed with 1% osmic acid for 3 h. Subsequently, the cells were dehydrated, embedded and stained. Finally, autophagosomes were observed by TEM (Philips, Amsterdam, The Netherlands).

Smad2 and Smad3 overexpression analysis

Plasmids containing smad2-pcDNA3.1(+) and smad3-pcDNA3.1(+) were designed and synthesized by General Bio Co., Ltd. (Anhui, China). HSC-T6 cells were prepared in dishes for 24 h, and 1.0 µg of plasmid/106 cells was transfected into HSC-T6 cells using Lipofectamine 2000. AKF-PD (2 mM) was added and incubated for 24 h. HSCs were detected by CCK-8 assays, Western blotting and acridine orange fluorescent staining.

Bioinformatics analysis

The transcriptome data of liver cancer patientswere obtained from The Cancer Genome Atlas (TCGA) database. The autophagy-related genes were collected from the Kyoto Encyclopedia of Genes and Genomes (KEGG) database. The box plot showing the expression differences of SMAD2 and SMAD3 in tumor tissues and normal tissues in TCGA was generated using the R package ggplot2. The survival curve regarding high SMAD2/SMAD3 and low SMAD2/SMAD3 expression in liver cancer patients in TCGA was generated using the R package survival. The univariate Cox regression analysis on SMAD2, SMAD3, age, gender, TNM staging system, tumor stage, and tumor grade in liver patients in TCGA was performed using the R package survival. The intercorrelations heatmap among autophagy-related genes in TCGA was generated using the R package ComplexHeatmap. The univariate Cox regression analysis on autophagy-related genes in TCGA was performed using the R package survival. The survival curve regarding high and low autophagy-related score groups in liver cancer patients in TCGA was generated using the R package survival. The correlation heatmap between SMAD2, SMAD3, and autophagy-related scores in liver cancer patients in TCGA was generated using the R package ComplexHeatmap. Gene set enrichment analysis (GSEA) of KEGG pathways on the differentially expressed genes (DEGs) between high and low autophagy-related score groups in liver cancer patients in TCGA was performed using the R package Pi. The drug prediction of autophagy-related scores based on the drug agents from the largest publicly available pharmacogenomics database Genomics of Drug Sensitivity in Cancer (GDSC) in liver cancer in TCGA was performed using the R package oncoPredict.

Statistical analysis

All experiments were repeated at least three times. GraphPad Prism 9 was used to analyse the data. The data are presented as the mean ± SD, and statistical significance was determined by one-way analysis of variance (ANOVA) with the post hoc test. P < 0.05 was considered statistically significant.

Results

AKF-PD attenuated CCl4-induced liver fibrosis in rats

In this study, a classic rat liver fibrosis model was created by intraperitoneal injection of CCl4 for 6 weeks. We observed the role of AKF-PD on liver injury by H&E staining. Intraperitoneal injection of CCl4 induced a large amount of hepatocyte necrosis, inflammatory cell invasion and fibre formation compared to the normal liver structure. However, AKF-PD and FFBJ (as a positive control group) evidently relieved hepatocyte injury and reduced fibrous scar areas (Fig. 1A). Liver fibrosis always occurs along with collagen accumulation. Masson staining and Sirius Red staining were applied to detect collagen content. Our results showed that in the CCl4 group, massive collagen was deposited in fibrotic liver scars, and collagen deposition was evidently decreased by AKF-PD and FFBJ treatment. However, there were no remarkable differences between the CCl4+AKF-PD group and the CCl4+FFBJ group. (P > 0.05) (Fig. 1A). Moreover, collagen I and collagen III , which are fibrotic markers, were measured in the rat liver. Western blotting showed that the expression of collagen I and III was observably increased by CCl4 treatment. However, AKF-PD and FFBJ therapy observably reduced their expression, but there were no remarkable differences between the two groups (P > 0.05) (Fig. 1B). The serum levels of ALT, TBIL, AST, and ALB were measured to examine hepatic function. Our results demonstrated that compared to the normal group, ALT, TBIL and AST levels were increased and ALB levels were decreased in the CCl4 group. However, AKF-PD treatment reversed these effects (Fig. 1C). Overall, our study indicated that AKF-PD could attenuate CCl4-induced liver fibrosis.

Figure 1 AKF-PD attenuated CCl4-induced liver fibrosis in rats.

A rat liver fibrosis model was established by intraperitoneal injection of CCl4 for 6 weeks. AKF-PD and FFBJ were intragastrically administered to the rats. (A) Liver sections were stained with H&E, Masson’s trichrome and Sirius Red (scale bar: 50 µm). (B) Western blot analysis of collagen I and collagen III protein expression in the livers. (C) The levels of ALT, AST, ALB and TBIL in rat serum were calculated. Values represent the mean ± SD (n = 6). * p < 0.05, ** p < 0.01. *** p < 0.001, N.S. not significant.

AKF-PD inhibited the TGF-β1/Smad pathway in CCl4-induced liver fibrosis

The TGF-β1/Smad pathway is an important pathway that promotes liver fibrosis. Western blot analysis revealed that TGF-β1, p-Smad2 and p-Smad3 expression in the CCl4 group was markedly increased and decreased by AKF-PD and FFBJ treatment. However, total Smad2 and Smad3 expression was not affected (Fig. 2A).

Figure 2 AKF-PD inhibited the TGF-β1/Smad signalling and autophagy in CCl4-induced liver fibrosis.

(A) Western blot analysis of TGF-β1, p-Smad2, Smad2, p-Smad3 and Smad3 protein expression in the liver. (B) Western blot analysis of Beclin-1, LC3-II /I and P62 protein expression in the livers. Values represent the mean ± SD (n = 6). * p < 0.05, ** p < 0.01. *** p < 0.001, N.S. not significant.

AKF-PD inhibited autophagy in CCl4-induced liver fibrosis

To study the influence of AKF-PD on autophagy, we examined the protein levels of Beclin-1, P62 and LC3, these are markers of autophagy. Western blot analysis revealed that the protein expression of Beclin-1 and LC3-II /I was remarkably upregulated in the CCl4 group; however, these levels were effectively reduced by AKF-PD and FFBJ therapy. In contrast, P62 protein expression was observably reduced in the CCl4 group and increased by AKF-PD and FFBJ therapy (Fig. 2B).

AKF-PD alleviated liver fibrosis by inhibiting the TGF-β1/Smad pathway in cultured HSCs

We further studied the impact of AKF-PD on the TGF-β1/Smad pathway in cultured HSCs. HSC activation and the expression of α-SMA, FN and collagen III are characteristic of liver fibrosis. Western blotting revealed that TGF-β1 stimulation increased α-SMA, FN and collagen III protein expression. Moreover, AKF-PD treatment observably reduced their expression in HSC-T6 cells. Furthermore, their expression levels were observably increased by the overexpression of Smad2 and Smad3 and reduced by treatment with AKF-PD (Fig. 3).

Figure 3 AKF-PD alleviated liver fibrosis by inhibiting the TGF-β1/Smad pathway in cultured HSCs.

Western blot analysis of α-SMA, FN and collagen III protein expression in HSCs. Values represent the mean ± SD of 3 independent experiments. * p < 0.05, ** p < 0.01. *** p < 0.001, N.S. not significant.

AKF-PD alleviated liver fibrosis by inhibiting HSC autophagy via the TGF-β1/Smad pathway in cultured HSCs

We researched the influence of AKF-PD on HSC autophagy in vitro. We analysed the viability of HSCs using a CCK-8 assay. Their viability at 48 h and 72 h was observably increased by TGF-β1 stimulation and inhibited by AKF-PD. Furthermore, the viability of HSC-T6 cells elevated dramatically in response to the overexpression of Smad2 and Smad3 and decreased after AKF-PD treatment (Fig. 4A). We researched the effect of AKF-PD on autophagy–lysosomes in HSCs by acridine orange fluorescent staining. The nucleus fluoresces green when acridine orange binds to double-stranded DNA, and red fluorescence occurs when it reacts with autophagy–lysosomes at low pH. We found that the area of red fluorescence was particularly clear in the TGF-β1 group. The area of red fluorescence was reduced and HSC-T6 cells were deformed in the TGF-β1+AKF-PD group. In addition, the area of red fluorescence was dramatically elevated by the overexpression of Smad2 and Smad3 and reduced after AKF-PD treatment. These results demonstrated that AKF-PD could decrease the number of autophagy–lysosomes in HSCs via the TGF-β1/Smad pathway (Fig. 4B). Electron microscopy was used to directly observe autophagosomes. The number of autophagosomes in HSC-T6 cells was observably increased by TGF-β1 stimulation and reduced by AKF-PD (Fig. 4C). In addition, Western blot analysis revealed that the protein expression of Beclin-1 and LC3-II /I in HSC-T6 cells was dramatically elevated by TGF-β1, and AKF-PD significantly reduced their expression. Moreover, P62 expression was dramatically reduced by TGF-β1 stimulation, and increased after treatment with AKF-PD. Furthermore, the expression of LC3-II /I and Beclin-1 was observably elevated by the overexpression of Smad2 and Smad3 and reduced after treatment with AKF-PD. Moreover, P62 expression was observably reduced by the overexpression of Smad2 and Smad3 and increased by treatment with AKF-PD. Thus, AKF-PD alleviated liver fibrosis by restraining HSC autophagy through the TGF-β1/Smad pathway (Fig. 4D). The mechanisms of how fluorofenidone alleviates liver fibrosis by inhibiting hepatic stellate cell autophagy via the TGF-β1/Smad pathway is shown in Fig. 5.

Figure 4 AKF-PD alleviated liver fibrosis by inhibiting HSC autophagy via the TGF-β1/Smad pathway in cultured HSCs.

(A) The viability of HSC-T6 cells was examined by CCK-8 assays. (B) Effect of AKF-PD on HSC autophagy, as determined by fluorescence microscopy after staining with acridine orange (scale bar: 50 µm). (C) Autophagosome structures (denoted by white triangles) in HSCs were obtained by TEM (scale bar: 0.5 µm). (D) Western blot analysis of Beclin-1, LC3-II /I and P62 protein expression in HSCs. Values represent the mean ± SD of 3 independent experiments. * p < 0.05, ** p < 0.01. *** p < 0.001, N.S. not significant.

Figure 5 Mechanism of AKF-PD treatment of liver fibrosis.

After liver injury caused by carbon tetrachloride, HSCs are activated and secrete TGF-β1. TGF-β1 combines with the type II receptor (TβRII) on the HSC membrane, triggering the phosphorylation of the TGF-β1 receptor (TβRI); subsequently, TβRI binds Smad2 and Smad3. This complex becomes phosphorylated and forms hetero-oligomers with Smad4; finally, they enter the nucleus to manage the transcription of target genes. These target genes include fibrosis-related genes, such as collagen I, α-SMA, MMPs, and TIMPs, and autophagy-related genes, such as Beclin-1. In this study, our findings demonstrated that AKF-PD reduced TGF-β1 secretion and inhibited TGF-β1/Smad signalling to inhibit HSC autophagy and reduce ECM synthesis, ultimately alleviating liver fibrosis.

The prognostic values of SMAD2 and SMAD3 in liver cancer

As liver fibrosis is closely connected to liver cancer, the pathogenic roles of SMAD2 and SMAD3 were further explored in liver cancer in TCGA. SMAD2 was significantly overexpressed in tumor tissues compared with normal tissues in TCGA (Fig. 6A). SMAD3 was significantly overexpressed in tumor tissues compared with normal tissues in TCGA (Fig. 6B). Liver cancer patients with high SMAD2 expression had significantly shortened survival times in TCGA (Fig. 6C). Liver cancer patients with high SMAD3 expression had significantly shortened survival times in TCGA (Fig. 6D). The univariate Cox regression analysis revealed that SMAD2, SMAD3, TNM staging system, and tumor stage were independent prognostic factors in liver patients in TCGA (Fig. 6E).

Figure 6 The prognostic values of SMAD2 and SMAD3 in liver cancer.

(A) The expression differences of SMAD2 in tumor tissues and normal tissues in TCGA. (B) The expression differences of SMAD3 in tumor tissues and normal tissues in TCGA. (C) The survival curve regarding high SMAD2 and low SMAD2 expression in liver cancer patients in TCGA. (D) The survival curve regarding high SMAD3 and low SMAD3 expression in liver cancer patients in TCGA. (E) The univariate Cox regression analysis on SMAD2, SMAD3, age, gender, TNM staging system, tumor stage, and tumor grade in liver patients in TCGA.

The development of autophagy-related scores in liver cancer

The pathogenic roles of autophagy were further explored in liver cancer in TCGA. The autophagy-related genes from the KEGG database were highly intercorrelated in TCGA (Fig. 7A). The univariate Cox regression analysis revealed autophagy-related genes, ATG12, ATG7, ATG3, ATG5, ATG4B, PIK3C3, and PRKAA2 were independent hazardous prognostic factors, while GABARAPL1 was favorable prognostic factor in liver patients in TCGA (Fig. 7B). The autophagy-related scores were developed using ssGSEA based on the prognostic autophagy-related genes. Liver cancer patients with high autophagy-related scores had significantly shortened survival times in TCGA (Fig. 7C).

Figure 7 The development of autophagy-related scores in liver cancer.

(A) The intercorrelations among autophagy-related genes in TCGA. (B) The univariate Cox regression analysis on autophagy-related genes in TCGA. (C) The survival curve regarding high and low autophagy-related score groups in liver cancer patients in TCGA.

The functional annotation of autophagy-related score in liver cancer

SMAD2 and SMAD3 significantly positively correlated with autophagy-related scores in liver cancer patients in TCGA (Fig. 8A). GSEA of KEGG pathways on the DEGs between high and low autophagy-related score groups in liver cancer patients in TCGA revealed that MAPK signaling pathway, cell cycle, mTOR signaling pathway, wnt signaling pathway, TGF-beta signaling pathway, VEGF signaling pathway, p53 signaling pathway, PI3K-Akt signaling pathway, and NOTCH signaling pathway were significantly enriched (Fig. 8B).

Figure 8 The functional annotation of autophagy-related score in liver cancer.

(A) The correlation between SMAD2, SMAD3, and autophagy-related scores in liver cancer patients in TCGA. (B) GSEA of KEGG pathways on the DEGs between high and low autophagy-related score groups in liver cancer patients in TCGA.

The drug prediction of autophagy-related score in liver cancer

The drug prediction of autophagy-related scores based on the drug agents from the GDSC database in liver cancer revealed that Mirin_1048, Palbociclib_1054, BMS-536924 1091, 1-BET-762 1624, OTX015 1626, AZD5153_1706, JAK 8517 1739, Cediranib 1922, GDC0B10_1925, I-BRD9 1928, UMI-77 1939, and BPD-00008900_1998 had significantly lower IC50 in the high autophagy-related score group (Fig. 9).

Figure 9 The drug prediction of autophagy-related score in liver cancer.

The drug prediction of autophagy-related scores based on the drug agents from the GDSC database in liver cancer.

Discussion

Liver fibrosis is a healing reaction of chronic liver injury. Hepatic inflammation activates HSCs, which leads to the synthesis and deposition of abundant ECM and finally forms liver fibrosis (Hernandez-Gea & Friedman, 2011; Zhou, Zhang & Qiao, 2014). The generation of a fibrous scar will lead to the destruction of the normal liver and the loss of liver cell function, eventually leading to liver failure (Kisseleva & Brenner, 2008). Except for the therapy of various causes of liver fibrosis, there is currently no effective treatment, but liver fibrosis is considered an invertible pathological process (Sun & Kisseleva, 2015). HSCs, which play a leading part in liver fibrosis, are crucial targets for antifibrotic treatment (Bataller & Brenner, 2001). Inhibiting HSCs activation is an important therapeutic approach to relieve liver fibrosis (Tao et al., 2013).

AKF-PD is a broad-spectrum antifibrotic drug that has significant effects on liver, kidney and lung fibrosis (Jiang et al., 2021; Tu et al., 2021; Liao et al., 2022). In this study, AKF-PD was applied to treat a CCl4-induced liver fibrosis model, which is a classic animal model for researching the mechanism of liver fibrosis (Jiang et al., 2019a). AKF-PD downregulated the levels of serum ALT, TBIL and AST and upregulated the levels of ALB, indicating that AKF-PD improved the liver damage caused by carbon tetrachloride and played a protective role in liver function. Furthermore, AKF-PD reduced the damage to hepatocytes and fibroplasia induced by carbon tetrachloride. In addition, AKF-PD dramatically lessened collagen I and III protein expression, which are the main ECM contents in liver cirrhosis (Gressner, 1995). In cultured HSCs, this study indicated that AKF-PD also inhibited α-SMA, FN and collagen III protein expression, which are activated HSC markers. We found that AKF-PD not only improved liver function but also improved CCl4-induced fibrosis in the liver.

Autophagy can decompose and recycle organelles, proteins and macromolecules in the cytoplasm to maintain homeostasis (Hazari et al., 2020). The formation of autophagosomes is the main feature of autophagy. Autophagy is regulated by autophagy-related genes (ATGs). LC3 promotes the formation of autophagosomes (Dikic & Elazar, 2018). The Beclin1 gene is a specific autophagy gene, and its upregulation can stimulate the occurrence of autophagy (Pan et al., 2015). p62 is an important autophagy protein, and its content is inversely correlated with the autophagy level (Liu et al., 2019). The activation of HSCs is related to autophagy, and several studies have shown that regulating autophagy in HSCs can effectively inhibit their activation and improve liver fibrosis (Hao et al., 2016; Park et al., 2021). We confirmed that AKF-PD could decrease the number of autophagosomes in HSCs. In addition, we detected the protein levels of LC3, P62 and Beclin-1 in the CCl4 fibrosis model and cultured HSC-T6 cells. AKF-PD inhibited the expression of Beclin-1 and LC3 and elevated P62. These results suggested that AKF-PD could inhibit autophagy in HSCs.

TGF-β1 is an intermediate substance that mediates fibrosis. TGF-β1 can promote the HSCs activation and the synthesis of ECM (Park et al., 2015). TGF-β1 combines with the type II receptor (TβRII) on the HSC membrane, triggering the phosphorylation of the TGF-β1 receptor (TβR I); subsequently, TβR I binds Smad2 and Smad3. This complex becomes phosphorylated and forms hetero-oligomers with Smad4; finally, they turn into the nucleus to manage the transcription of target genes (Inagaki & Okazaki, 2007; Hernandez-Gea & Friedman, 2011).These target genes include fibrosis related genes, such as collagen I, α-SMA, MMPs, and TIMPs (Inagaki & Okazaki, 2007; Liu et al., 2019), and autophagy-related genes, such as Beclin-1 (Pan et al., 2015). The TGF-β1/Smad pathway can promote ECM synthesis and regulate HSC autophagy during liver fibrosis (Wu et al., 2017; Ma et al., 2019). The TGF-β1/smad pathway is a signaling pathway that plays a crucial role in various cellular processes, including cell growth, differentiation, and apoptosis. It is activated by the binding of TGF-β1 (transforming growth factor-beta) to its receptor, which leads to the phosphorylation and activation of Smad proteins. Smad2 and Smad3 are two key members of the Smad family of proteins that are involved in the TGF-β1/smad pathway. Once activated, Smad2 and Smad3 form a complex with Smad4 and translocate into the nucleus, where they regulate the transcription of target genes. In liver cancer, the TGF-β1/smad pathway is often dysregulated, leading to abnormal cell growth and tumor progression. Smad2 and Smad3 have been found to have both tumor-suppressive and tumor-promoting effects in liver cancer, depending on the context. They can inhibit cell proliferation, induce cell cycle arrest, and promote apoptosis, thereby acting as tumor suppressors. On the other hand, they can also promote tumor invasion, metastasis, and angiogenesis, acting as tumor promoters. The exact mechanisms by which Smad2 and Smad3 exert their effects in liver cancer are still being studied. However, it is believed that they interact with other signaling pathways and transcription factors to regulate the expression of genes involved in cell growth, survival, and metastasis. It is important to note that the TGF-β1/smad pathway and the roles of Smad2 and Smad3 in liver cancer are complex and multifaceted, and further research is needed to fully understand their mechanisms of action and potential therapeutic implications.

In our study, we discovered that AKF-PD decreased the protein expression of TGF-β1, p-Smad2 and p-Smad3 in the CCl4-induced rat liver fibrosis model. These results indicated that AKF-PD can inhibit the TGF-β1/Smad pathway. Furthermore, the viability and the number of autophagosomes in cultured HSCs were observably increased by the overexpression of Smad2 and Smad3 and reduced by AKF-PD. Moreover, the expression of Beclin-1 and LC3 were observably elevated and that of P62 was reduced by overexpression of Smad2 and Smad3, and AKF-PD reversed these effects. Moreover, the expression of Beclin-1 and LC3 was observably elevated and that of P62 was reduced by overexpression of Smad2 and Smad3, and AKF-PD reversed these effects. These results indicated that AKF-PD inhibited the TGF-β1/Smad pathway to suppress HSC autophagy. In addition, FN, α-SMA and collagen III expression was observably elevated by overexpression of Smad2 and Smad3 and was reduced by AKF-PD treatment. This study confirmed that AKF-PD reduced TGF-β1 secretion and inhibited the TGF-β1/Smad signalling pathway to inhibit HSC autophagy and reduce ECM synthesis, ultimately alleviating liver fibrosis.

By modifying the tumor milieu, fibrosis can affect how liver cancer develops (Affo, Yu & Schwabe, 2017). In addition to changing the mechanical characteristics of the liver, ECM deposition in the fibrotic liver also has the power to coordinate and modulate numerous signaling networks in epithelial, endothelial, stromal, immune, and tumor cells. It does this by either directly binding specific receptors, like integrins and growth factor receptors, or by forming complexes with ligands that increase their activity and promote binding to their receptors, which promotes the growth and survival of transformed tumor cells (Baglieri, Brenner & Kisseleva, 2019).

HSCs contribute to the growth of liver carcinoma in both direct and indirect ways. Senescence of HSCs as well as HSC stimulation, could have a significant impact on how a tumor develops. By secreting vital cytokines and chemokines like HGF, TGF-β, PDGF, IL-6, and Wnt ligands that can directly affect the tumor cells, HSC activation can also directly impact the formation of liver cancer (Dhar et al., 2020) .

In our study, SMAD2 and SMAD3 were hazardous factors in liver cancer. SMAD2 and SMAD3 correlated with autophagy-related scores (based on ATG12, ATG7, ATG3, ATG5, ATG4B, PIK3C3, PRKAA2, and GABARAPL1) in liver cancer. A similar approach was also mentioned in the published study (Wang et al., 2020; Kang et al., 2021). Increased ATG5-ATG12 and their involvement in apoptosis in liver cancer were reported (Kunanopparat et al., 2016) . ATG7 knockdown prevents tumorigenesis in a mouse liver cancer model (Cho et al., 2021). In liver cancer, lncRNA NEAT1 increases autophagy by controlling miR-204/ATG3 and increases cell resilience to sorafenib (Li et al., 2020). Cancer cells with ATG4B knockouts have suppressed autophagy and activated AMPK for cell cycle arrest (Liu et al., 2017). PIK3C3 controls the growth of liver cancer stem cells, and PIK3C3 inhibition inhibits the action of liver cancer stem cells brought on by PI3K inhibitor (Liu et al., 2020). In liver cancer, loss of GABARAPL1 gives ferroptosis resistance to cancer stem-like cells (Du et al., 2022). Autophagy-related scores could predict drug response in liver cancer, which Mirin_1048, Palbociclib_1054, BMS-536924 1091, 1-BET-762 1624, OTX015 1626, AZD5153_1706, JAK 8517 1739, Cediranib 1922, GDC0B10_1925, I-BRD9 1928, UMI-77 1939, and BPD-00008900_1998 had significantly lower IC50 in the high autophagy-related score group. Our study provided some implications about how liver fibrosis was connected with liver cancer by SMAD2/SMAD3 and autophagy. Our study could help with the clinical management of liver fibrosis and liver cancer.

Conclusion

In summary, our study proves that the inhibition of HSC autophagy plays an important part in the antifibrotic effects of AKF-PD. AKF-PD inhibits HSC autophagy, at least partly, through the TGF-β1/Smad signalling pathway.

Supplemental Information

Supplemental Information 1 The source data for Fig. 1.

Click here for additional data file.

Supplemental Information 2 The source data for Fig. 2.

Click here for additional data file.

Supplemental Information 3 The source data for Fig. 3.

Click here for additional data file.

Supplemental Information 4 The source data for Fig. 4.

Click here for additional data file.

Supplemental Information 5 The source data for Fig. 5.

Click here for additional data file.

Supplemental Information 6 The source data for Bioinformatics analysis in Figs. 6–9.

Click here for additional data file.

Supplemental Information 7 ARRIVE 2.0 Checklist.

Click here for additional data file.

Additional Information and Declarations

Competing Interests

Author Contributions

Animal Ethics

Data availability

The authors declare that there are no competing interests.

Xiongqun Peng conceived and designed the experiments, performed the experiments, analyzed the data, prepared figures and/or tables, authored or reviewed drafts of the article, and approved the final draft.

Huixiang Yang conceived and designed the experiments, performed the experiments, analyzed the data, prepared figures and/or tables, authored or reviewed drafts of the article, and approved the final draft.

Lijian Tao analyzed the data, authored or reviewed drafts of the article, and approved the final draft.

Jingni Xiao conceived and designed the experiments, performed the experiments, analyzed the data, prepared figures and/or tables, authored or reviewed drafts of the article, and approved the final draft.

Ya Zeng analyzed the data, authored or reviewed drafts of the article, and approved the final draft.

Yueming Shen conceived and designed the experiments, performed the experiments, analyzed the data, prepared figures and/or tables, authored or reviewed drafts of the article, and approved the final draft.

Xueke Yu conceived and designed the experiments, performed the experiments, analyzed the data, prepared figures and/or tables, authored or reviewed drafts of the article, and approved the final draft.

Fei Zhu conceived and designed the experiments, performed the experiments, analyzed the data, prepared figures and/or tables, authored or reviewed drafts of the article, and approved the final draft.

Jiao Qin conceived and designed the experiments, performed the experiments, analyzed the data, prepared figures and/or tables, authored or reviewed drafts of the article, and approved the final draft.

The following information was supplied relating to ethical approvals (i.e., approving body and any reference numbers):

The Ethics Review Committee for Animal Experimentation of University of South China approved this study.

The following information was supplied regarding data availability:

The raw data of the vitro/in vivo experiments and all bioinformatics analysis are available in the Supplementary Files.

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
