# Peer review of "Fluorofenidone alleviates liver fibrosis by inhibiting hepatic stellate cell autophagy via the TGF-β1/Smad pathway: implications for liver cancer"

_PeerJ, doi:10.7717/peerj.16060_

## Round 0.1 · original submission · Major Revisions

Please revise the manuscript as the reviewers suggested.

Reviewer 1 ·

Basic reporting

The study convincingly demonstrates the significant role of inhibiting hepatic stellate cell (HSC) autophagy in the antifibrotic effects of AKF-PD. The authors provide evidence that AKF-PD effectively inhibits HSC autophagy, partially mediated by the TGF-β1/Smad signaling pathway. Additionally, the study explores the prognostic value of SMAD2 and SMAD3 in liver cancer. While the research is well-designed and informative, there are some areas that require attention. These issues should be addressed to improve the overall quality and clarity of the manuscript.

Reviewer Comments:
1) The discussion section lacks logical flow, and the transition between the study of liver fibrosis and liver cancer needs improvement. Ensure that the discussion is organized in a coherent manner, clearly presenting the findings related to liver fibrosis and smoothly transitioning to the implications for liver cancer. This will help readers understand the interconnectedness between these topics and their relevance to the study.

2) Rewrite the figure legends for Figures 6-9 to enhance clarity. Provide concise and informative descriptions of each figure, clearly stating what is being presented and the key findings or observations. This will ensure that readers can easily understand the content of the figures without confusion.

3) Conduct a function annotation of SMAD2 and SMAD3 to provide more insights into their roles in liver fibrosis and liver cancer. Include a detailed analysis of their known functions, pathways they are involved in, and their specific contributions to the pathogenesis of these diseases. This will add depth to the understanding of the mechanisms underlying the observed effects of AKF-PD.

4) In the introduction section, briefly introduce the implications of liver fibrosis for liver cancer. Provide a concise overview of how liver fibrosis serves as a critical stage in the progression towards liver cancer, highlighting the importance of addressing liver fibrosis as a potential therapeutic target to prevent or delay the development of liver cancer. This will set the stage for the relevance of the study and its contribution to the field.

5) Seek assistance from a native English speaker for English editing. It is important to ensure that the manuscript is grammatically correct, uses appropriate terminology, and reads smoothly. A thorough proofreading by a language expert will enhance the clarity and overall quality of the manuscript, making it more accessible to a wider audience.

Experimental design

Same as above.

Validity of the findings

Same as above.

Reviewer 2 ·

Basic reporting

English needs to be improved.

Experimental design

1, The introduction is a mess. Maybe the authors should rearrange the introduction as follow: liver fibrosis may induce liver cancer, the current medication for liver fibrosis (both Western and Chinese medicine), what is the advantage of AKF-PD, what is unclear of AKF-PD and maybe autophagy plays an essential role.
2, [Line 86 to 89] Peng Y et al. had published so many studies on AKF-PD in animal and HSC (reference 8, 20 and 21). Why the authors still think the mechanism is not completely clear? The patterns you use and the signal pathway you found were kept in line with the previous ones, so what is your highlight?
3, [2.1] Why chose the male rats? Why use full-fat diet? How many rats in each group? How to calculate the sample size? How is the drug dose determined? Why intervention for 6 weeks? The sodium pentobarbitone is forbidden in mainland China, how did you get those legally? The liver tissue for staining and Western blotting were different, please specify the each procedure.
4, [2.2] The merchandise numbers were missing for each commercial kit.
5, [2.7] The protein and its phosphorylated protein were developed in the same membrane or developed in two membranes?
6, [2.11] I cannot find any data presented as the mean ± SD throughout your manuscript. Did you test the normal distribution?
7, The authors mentioned AMPK-mTOR signal pathway in Introduction and annotated in 3.8. Why did not test the protein expression of this pathway?
8, [Line 3.6 to 3.9] Why provide these results? They are not relevant with AKF-PD and liver fibrosis at all. Therefore, your conclusion did not mention these results at all.
9, [Line 294 to 299] Maybe the “3.9” is missing.

Validity of the findings

Although with great efforts had been made by all the authors, the scientific significance of this manuscript is not very obvious.

---

## Round 0.2 · accepted · Accept

This manuscript can be accepted now.

Reviewer 1 ·

Basic reporting

The author perfectly solved all my doubts, and I recommend accepting this manuscript.

Experimental design

The author perfectly solved all my doubts, and I recommend accepting this manuscript.

Validity of the findings

The author perfectly solved all my doubts, and I recommend accepting this manuscript.

Additional comments

The author perfectly solved all my doubts, and I recommend accepting this manuscript.

Reviewer 2 ·

Basic reporting

Well done.
Please add those methods in the manuscript (e.g. statistical methods), but not only presented to the Reviewers.

Experimental design

no comment

Validity of the findings

no comment

Additional comments

Please provide the full name of "FN" in the Abstract.